# Three-dimensional structural dynamics of DNA origami Bennett linkages using individual-particle electron tomography

Dongsheng Lei[1], Alexander E. Marras [2], Jianfang Liu[1], Chao-Min Huang[2], Lifeng Zhou[2], Carlos E. Castro [2], Hai-Jun Su[2] & Gang Ren[1]

Scaffolded DNA origami has proven to be a powerful and efficient technique to fabricate functional nanomachines by programming the folding of a single-stranded DNA template strand into three-dimensional (3D) nanostructures, designed to be precisely motion-controlled. Although two-dimensional (2D) imaging of DNA nanomachines using transmission electron microscopy and atomic force microscopy suggested these nanomachines are dynamic in 3D, geometric analysis based on 2D imaging was insufficient to uncover the exact motion in 3D. Here we use the individual-particle electron tomography method and reconstruct 129 density maps from 129 individual DNA origami Bennett linkage mechanisms at ~ 6–14 nm resolution. The statistical analyses of these conformations lead to understanding the 3D structural dynamics of Bennett linkage mechanisms. Moreover, our effort provides experimental verification of a theoretical kinematics model of DNA origami, which can be used as feedback to improve the design and control of motion via optimized DNA sequences and routing.

[1] The Molecular Foundry, Lawrence Berkeley National Laboratory, Berkeley, CA 94720, USA. [2] Department of Mechanical and Aerospace Engineering, The Ohio State University, Columbus, OH 43210, USA. Correspondence and requests for materials should be addressed to H.-J.S. (email: su.298@osu.edu) or to G.R. (email: gren@lbl.gov)

The development of functional nanomachines paves the way for the advancement of novel devices and processes. Nanomachines have demonstrated a potential to revolutionize medicine, and reduce resource consumption and environmental pollution in manufacturing processes. They also enable new manufacturing processes to assemble objects, such as biomolecules, gold nanoparticles, quantum dots, and polymers, in a precise and efficient way. Scaffolded DNA origami[1] has proven to be a powerful and efficient technique to fabricate nanostructures by programming the folding of a single-stranded DNA viral genome (called the scaffold, ~ 8,000 bases) into three-dimensional (3D) nanostructures using multiple shorter synthetic single-stranded DNA (called staples, ~ 30 bases long). The folding occurs in a molecular self-assembly process utilizing Watson–Crick DNA base pairing.

Recently, Castro and colleagues[2,3] have demonstrated that the nanotechnology can be used to design and fabricate DNA origami mechanisms (DOM) with multiple degrees of freedom and complex 2D and 3D motion. DOM can exhibit motions, which can be actuated using DNA strands as an input. This actuation can be reversed using DNA strand displacement[4], providing precise motion control. To advance the analysis and future design of dynamic DOM, a thorough understanding of the 3D structure and motion of DOM is required.

In earlier works, Castro and colleagues[2] used transmission electron microscopy (TEM) to image DOM deposited on a surface and obtained geometric parameters, including lengths and angles. As these measurements were performed only on two-dimensional (2D) projections, they are inadequate for complete validation of 3D structures. Furthermore, although single-particle 3D reconstruction methods using TEM can achieve atomic resolution, it requires image averaging over many homogeneous structures. This is unsuitable for dynamic mechanisms that adopt many different conformations. For these two reasons, individual-particle electron tomography (IPET) was chosen to study the 3D structures of each individual DOM (without averaging from different DOM molecules). The IPET reconstruction method requires no pre-given initial model, class averaging, or lattice, but can tolerate small tilt errors and large-scale image distortion to achieve an intermediate resolution of 3D reconstruction (1–2 nm for negative staining samples and 3–5 nm for cryo-electron microscopy (cryo-EM) samples) via decreasing the reconstruction image size[5]. In the IPET iterative refinement process, automatically generated dynamic low-pass filters and soft masks were sequentially used. The IPET 3D reconstruction approach has been used to study the 3D structure and dynamics of DNA-nanogold conjugations[6], very low density human lipoproteins[7], immunoglobulin-G1 antibodies[8], and many other macromolecules[9–12].

In this study, we used the IPET method[5] to reconstruct the 3D structure from a tilt series of each individual DOM. By comparing the conformations revealed from each 3D reconstruction, this approach enables us to understand how the underlying design is related to the structural mobility and motion of DNA origami-based machines.

## Results

**Electron microscopic images of DNA origami Bennett linkages.** Samples of DNA origami Bennett linkages (Fig. 1a) were prepared by three sample preparation methods, i.e., optimized negative-staining (OpNS),[13–15] cryo-EM, and cryo-positive staining electron microscopy method (cryo-PS).[16]

The survey of the OpNS-EM micrograph of DNA origami Bennett linkage showed evenly distributed particles (Fig. 1b and Supplementary Fig. 1). The micrographs were bandpass filtered between 0.0066 and 1 nm$^{-1}$. The representative particles showed a "quadrilateral" shape (Fig. 1c). Further examination indicates

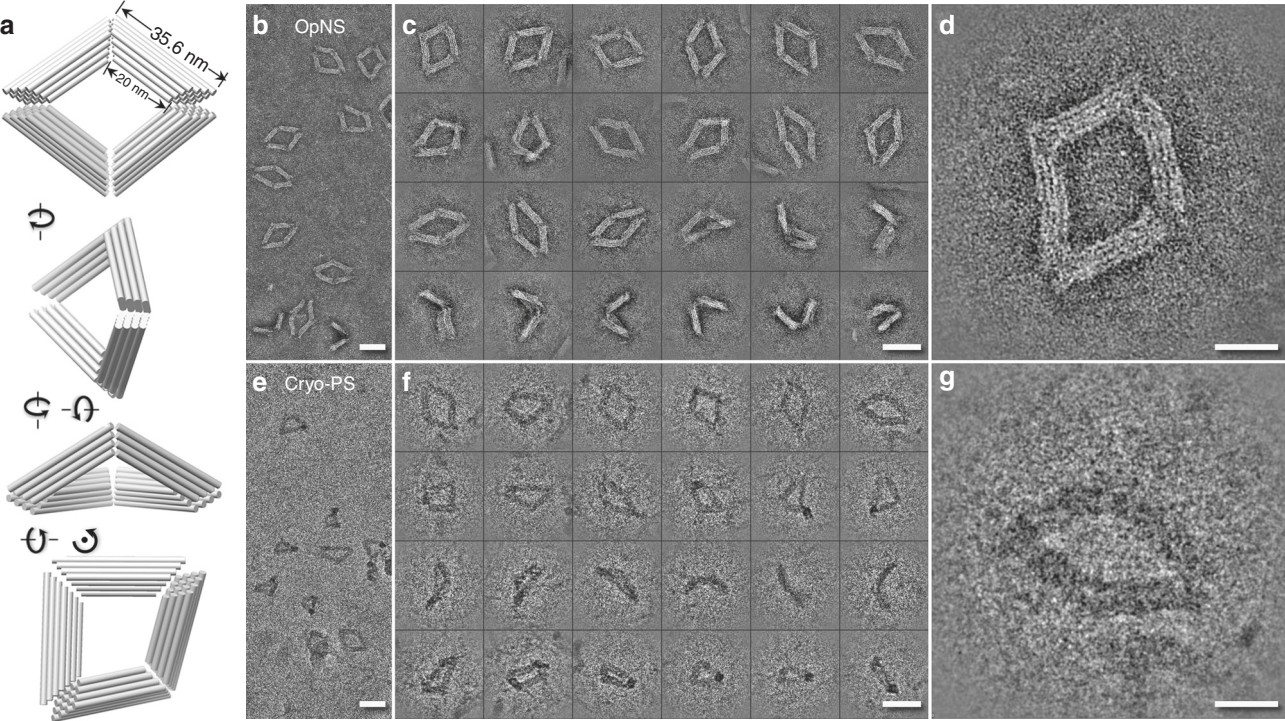

**Fig. 1** Schematic and EM images of DNA origami Bennett linkage. **a** A schematic introducing the model of Bennett linkage. **b** A survey OpNS image. **c** Twenty-four representative particles by OpNS. **d** A zoom-in OpNS image. **e** A survey cryo-PS image. **f** Twenty-four representative particles by cryo-PS. **g** A zoom-in cryo-PS image. Scale bars are 50 nm in **b**, **c**, **e** and **f**, and are 20 nm in **d** and **g**

that the particles vary both in shape and conformation, including a square-like "open" conformation and a compacted, bundle-like, "closed" conformation (Fig. 1c). These images are consistent with a previously reported observation[2], thus providing a new line of evidence that the DNA origami Bennett linkages are structurally flexible and exhibit heterogeneous conformations. The larger OpNS-EM images show that the four arms of the quadrilateral-shaped particles have similar widths ranging from 9.5 to 10.5 nm, and each arm is composed of four visible layers (Fig. 1c). The similarity in width and internal structure of arms suggests that the DNA origami Bennett linkages had been well-assembled. However, some of the arms have a curved shape (Fig. 1d), which was not predicted in design, suggesting the structures have substantial flexibility. This observation was consistent with that from computational predictions[17] and simulations[18]. Moreover, the distinct shape of arms and overall particle shapes suggest those images are useful for revealing the basic structure of a DNA origami Bennett linkage.

Although OpNS technique provides high-contrast images with detailed structure, potential artifacts, such as a preferred particle orientation to the supporting film, flatness during drying, and the staining reaction of the particle, may influence the determination of the 3D structure. Cryo-EM is an ideal method to study protein structures under near-native conditions. However, the contrast of cryo-EM images of the Bennett linkages was too low to be visualized directly, which made it difficult to use for validation of the final 3D reconstruction. To have a sufficient image contrast, we used our reported cryo-PS method to prepare the sample[16]. By this method, the holey carbon films were used to support a piece

of thin film of amorphous ice across the hole areas. The ice-crossed holes could prevent potential artifacts from the particle directly attaching to the substrate.

The survey of cryo-PS micrographs showed that DNA origami Bennett linkages were clearly visible (Fig. 1e, after bandpass filtering between 0.0066 to 1 nm$^{-1}$). The representative particle images (Fig. 1f) confirmed the quadrilateral-shaped particles ranging from "open" to "closed" conformations similar to those observed by OpNS (Fig. 1c). However, particles showed a higher diversity in shape than those from OpNS, suggesting the particles have less influence on their conformations by the supporting carbon film. However, the slightly curved arms (Fig. 1g) confirmed the flexibility of DNA bundles observed from the OpNS and further supported the prediction based on the computation and molecular dynamic simulations[17,18]. Moreover, the high contrast of images with less influence on conformation or flatness makes it possible to perform 3D reconstruction from each individual DNA origami Bennett linkage particle.

**IPET 3D reconstruction of two DNA origami Bennett linkages.** Owing to the structural heterogeneity, each DNA origami Bennett linkage particle has a unique 3D structure and conformation. Thus, any averaging method on the heterogeneous particles was not suitable for analysis of the 3D structure and overall motion. Here we used our reported IPET method[5] to obtain a 3D structure from each individual particle of the Bennett linkages.

For IPET 3D reconstruction, the cryo-PS samples of the Bennett linkages were imaged and underwent a series of tilt angles from −45° to 45° at a 1.5° increment (Fig. 2a). The tilt

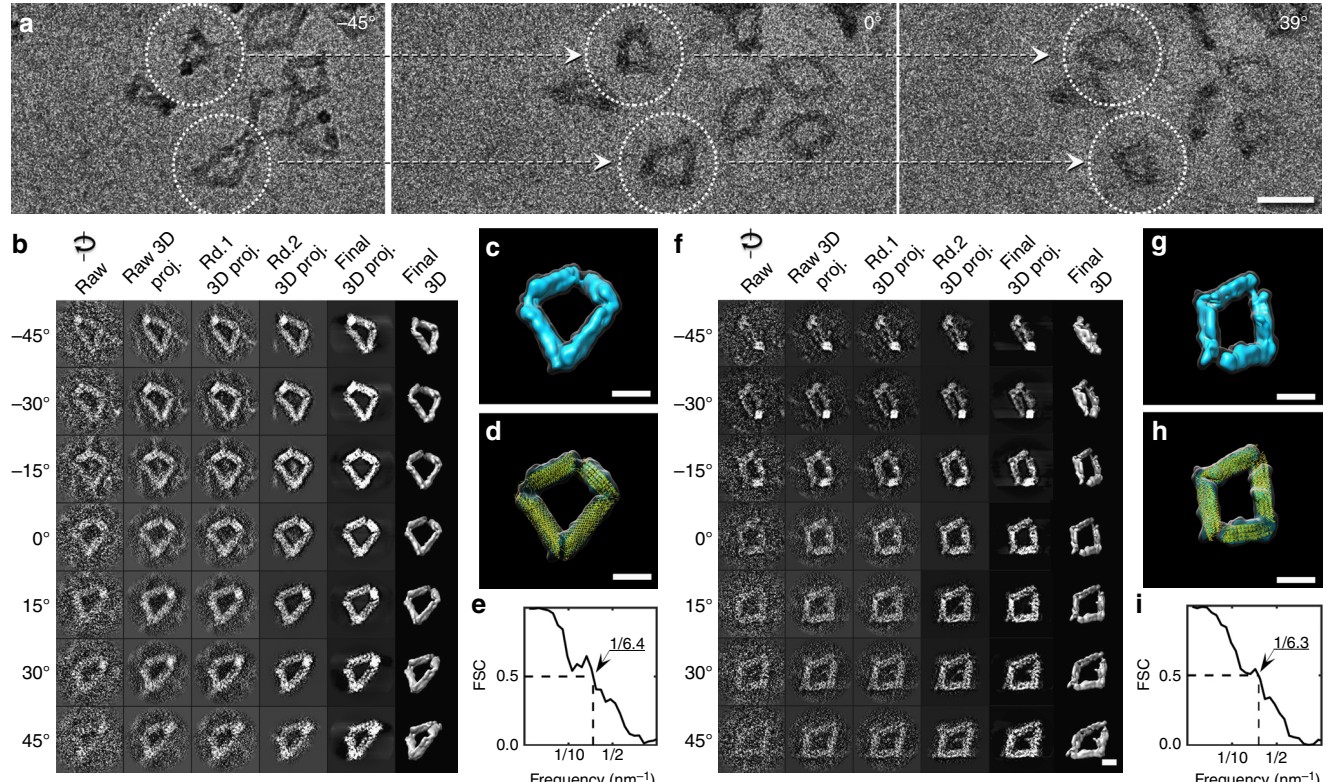

**Fig. 2** IPET 3D reconstruction of two DNA origami Bennett linkages. **a** Representative cryo-PS tilt series, in which targeted particles (white circles) are linked by dashed arrows. **b** IPET 3D reconstruction process. Seven representative tilt images of a targeted particle (first column) were gradually aligned via an iterative refinement process (next five columns). **c** Final IPET 3D map. **d** Flexibly docked with the DNA origami Bennett linkage model. **e** The FSC analyses showed the resolution was ~ 6.4 nm. **f–i** The IPET 3D reconstruction of another targeted Bennett linkage with a resolution of ~ 6.3 nm. Scale bars are 50 nm in **a** and 20 nm in **b–d** and **f–h**

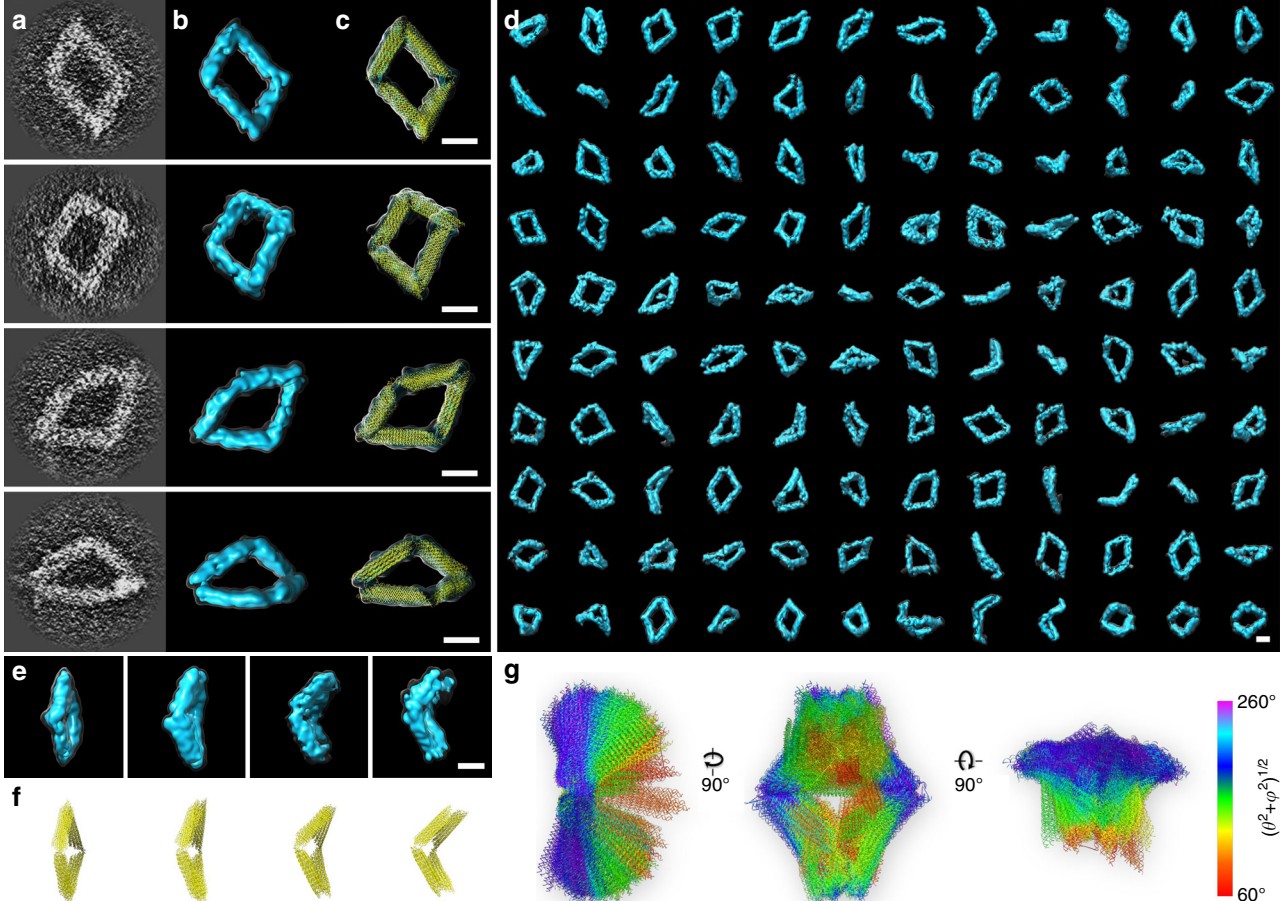

**Fig. 3** 3D structures and conformations of 129 particles of DNA origami Bennett linkage. **a** The projections of four representative IPET 3D reconstructions (with circular mask only). **b** Final 3D reconstructions. **c** Flexibly docked with the model (yellow ribbon) into the envelopes of final 3Ds. **d** 120 additional IPET 3Ds from 120 individual particles (orientations are same as that in specimen). **e** Four representative particles with different conformations. **f** The corresponding models. **g** All 129 models aligned to each other and colorized based on the root sum square of angles (angle definition shown in Fig. 4). Scale bars are 20 nm

images of each targeted particles were iteratively aligned to a global center to achieve a final ab initio 3D reconstruction. Only particles whose overall shape was visible through the entire tilt range were selected. The step-by-step refinement procedure and the intermediate results are shown in Fig. 2b. The final 3D density map was achieved at a 3D resolution of ~ 6.4 nm based on Fourier shell correlation (FSC) and a criterion of 0.5 (Fig. 2e; details are provided in the Methods section). After low-pass filtering to 8.0 nm, the final 3D density map displayed in Fig. 2c showed a quadrilateral-shaped structure with an overall dimension of ~ 70 nm and arm lengths of ~ 40 nm. The overall shape confirmed the success of the designed structure[2].

Although the resolution of the reconstructed map was insufficient to determine the secondary structure of this Bennett linkage, the overall map can be used as a constraint to flexibly dock an initial model of the Bennett linkage, built using CanDo,[17,19] to obtain a new Bennett linkage conformation. By satisfying both the best overall fit to the density map and a minimal overlap among arms within the model, we changed the joint angles between two adjoined arms (Fig. 2d) to obtain a new conformation of the Bennett linkage that differed from the original model (Fig. 2d).

By repeating the above process on another targeted particle, we reconstructed a second 3D map for a Bennett linkage (Fig. 2f–i). The representative tilt images showed that this Bennett linkage was also visible in each tilt image. Through IPET reconstruction,

the final 3D reconstruction at ~ 6.3 nm resolution showed that this particle also has a quadrilateral shape like the first one. However, it exhibited less bending angles between two nearby arms (Fig. 2g). By flexible docking of the Bennett linkage model into this map, a second conformation of a Bennett linkage was obtained (Fig. 2h).

**IPET 3Ds of 127 more DNA origami Bennett linkages.** Through particle-by-particle 3D reconstructions using IPET, another 127 particles of Bennett linkages were targeted from a pool of ~ 200 particles in 20 sets of tilt series (Fig. 3a–d and Supplementary Fig. 2 to 66). These 127 density maps, with resolutions ranging from 5.6 to 13.9 nm, confirmed the overall quadrilateral shape, ranging from a square-like open conformation to a more compacted conformation (Fig. 3e, f). Although each DNA helix within the Bennett linkages could not be identified under the current resolution, the spatial location and orientation of each arm of Bennett linkages can be defined based on the 3D density maps. As a result, 127 more conformations were revealed (Fig. 3g, Supplementary Fig. 2 to 66, and Supplementary Table 1).

**Statistical analyses of the conformations.** Aligning all the 129 conformations of DNA origami Bennett linkages shows a variety of conformations (Fig. 3g). To quantitatively analyze the conformational dynamics and fluctuations, two sets of bending angles

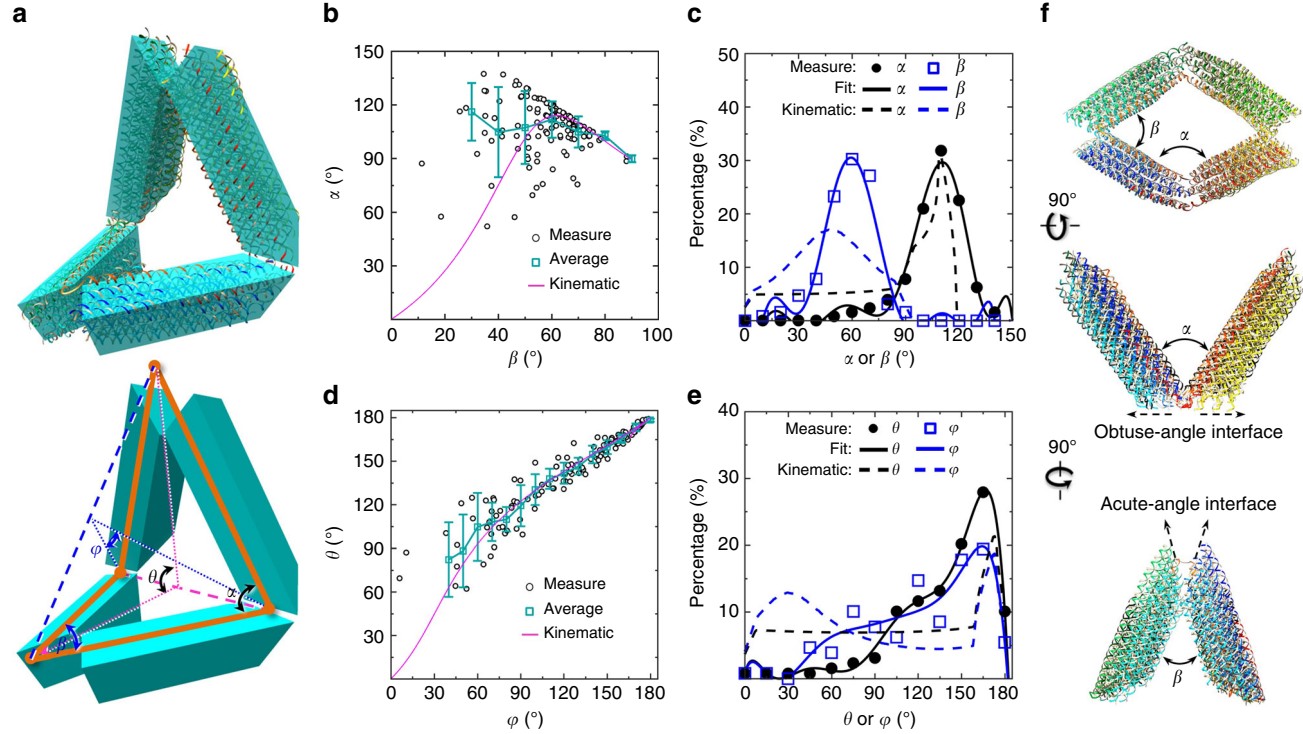

**Fig. 4** Angle distributions of the 129 particles of the DNA origami Bennett linkage. **a** Schematic shows the definition of Bennett linage internal angles $\alpha$ and $\beta$ (between two adjacent arms), and the dihedral angles $\theta$ and $\varphi$. **b** Distribution of internal angle $\alpha$ against $\beta$. Averaged $\alpha$ within the angle $\beta$ range of 10° are displayed by cyan line (the error bars show the SD). The pink curve was predicted by using kinematic analysis. **c** The histogram of internal angles $\alpha$ and $\beta$ was normalized and then fitted using a sixth degree polynomial function, in which, R-factor is 0.992 and STD is 1.28 for $\alpha$, and R-factor is 0.976 and STD is 3.90 for $\beta$. **d** Distribution of the dihedral angle $\theta$ against $\varphi$. Averaged $\theta$ within the angle $\varphi$ range of 10° are displayed by cyan line (the error bars show the SD). The pink curve was predicted by using kinematic analysis. **e** The histogram of dihedral angles $\theta$ and $\varphi$ was normalized and then fitted using a sixth degree polynomial function, in which, R-factor is 0.997 and STD is 1.36 for $\theta$, and R-factor is 0.937 and STD is 4.42 for $\varphi$. **f** Three orientations of the model showed the connection regions between two adjacent arms that formed the internal angle $\alpha$ and $\beta$

formed within the quadrilateral-shape Bennett linkages were measured, i.e., the internal angles $\alpha$ and $\beta$ that were defined as the angles between two adjacent arms, and the dihedral angles $\theta$ and $\varphi$ that were defined as the angles between two triangles formed by two adjacent arms (Fig. 4a). Considering the resolution of 3D maps was insufficient to identify the different arms, the measured larger angle was defined as angle $\alpha$, and the smaller angle was defined as $\beta$. Similarly, the larger dihedral angle was defined as angle $\theta$, and the smaller dihedral angle was defined as angle $\varphi$. Both sets of angles determined the overall conformation of Bennett linkages. For example, when $\alpha$ and $\beta$ were small, Bennett linkages underwent a compacted conformation. In contrast, when $\alpha$ and $\beta$ were large, Bennett linkages formed a flat and square-like conformation. To study the conformations, the root sum squared internal angles $\theta$ and $\varphi$ was computed and used as a single degree of freedom that defines the conformation from the "open" to "closed" states (Fig. 3g).

The statistical analysis of the internal angle $\alpha$ against $\beta$ showed two angles were near randomly distributed within a narrow angle range, in which more than 80% of the data were distributed with the $\alpha$ angle ranging from ~ 50 ° to ~ 130 °, and $\beta$ angle ranging from ~ 15 ° to ~ 90 ° (Fig. 4b), suggesting the internal angles were distinct from each other. The distribution of the averaged angle $\alpha$ (within an angle range of 10°) showed a poor agreement with the kinematics model[2,20] (model detail is provided in the Supporting Information and Supplementary Fig. 67). More than ~ 80% of data distributed within the narrow angle range reflected a low energy well of the conformations within the angle ranges of ~ 50° to ~ 130° for $\alpha$ and ~ 15 ° to ~ 90° for $\beta$.

Both histograms of the percentage of angle $\alpha$ and $\beta$ showed a near Gaussian distribution (Fig. 4c) with R-factor of 0.99 and 0.98, respectively (after fitting with a sixth-degree polynomial function). The peaks of the distributions were similar to that of the kinematic prediction,[2,20] especially for the angle $\alpha$. However, the shape of the distributions was different from the plot of the kinematics equation, especially for angle $\beta$, in which the distribution from the model was significantly wider and lower, with experiments suggesting a deeper energy well for angle $\beta$ around 60°.

The statistical analysis of the dihedral angles of $\theta$ against $\varphi$ showed a near linear distribution with R-factor of 0.93, suggesting a correlation between $\theta$ and $\varphi$ (Fig. 4d). The averages of $\theta$, within a sampling angle of 10° of angle $\varphi$, against angle $\varphi$ showed near perfect match to the plot based on the kinematics equation (purple line in Fig. 4d) within the range of 60°–180° for $\theta$ and 40°–180° for $\varphi$ angle. The linear distribution of $\theta$ against $\varphi$ suggests a certain level of correlation between $\theta$ and $\varphi$. The angles rarely appear within the range of 0°–60° for $\theta$ and 0°–40° for $\varphi$, suggesting a high energy barrier around these angle ranges.

The histograms of the percentages of angle $\theta$ and $\varphi$ both have a dominant peak at ~ 130°–160°, revealing the low energy well of the conformations near the "open" conformation (Fig. 4e). Moreover, the similar shape of both curves further confirmed the correlation between $\theta$ and $\varphi$. In contrast, the percentages from kinematic analysis were not like each other or the experimental curves, suggesting the freedom or energies of the conformations were more unevenly distributed than predicted.

## Discussion

Scaffolded DNA origami has proven to be a powerful and efficient technique to fabricate nanostructures. The understanding of 3D structures and dynamics of fabricated DNA origami would pave the way to devise more precise and efficient manufacturing processes and functional designs. The 2D imaging of fabricated DOMs using TEM suggested those mechanisms are dynamic in 3D structure[2]. To validate that fabricated DOMs have 3D conformations as designed, our efforts were devoted to the 3D structure of an example DOM, i.e., Bennett linkage.

Although the 3D structure of rigid DNA origami has been obtained at ~ 1 nm resolution using cryo-EM and single-particle reconstruction methods[21], to best of our knowledge, 3D structure of individual flexible DNA origami has not yet been achieved. The heterogeneity of DNA origami Bennett linkage makes it impractical to study by means of NMR or single-particle EM studies that include averaging over many particles. Recently cryo-electron tomography was used to study the structural diversity of supercoiled DNA[22]. We used IPET and cryo-PS methods to reconstruct 3D maps of 129 individual DNA origami Bennett linkages. Although the resolution of obtained maps (no more than ~ 5 nm) was relatively low compared with the resolutions reported for rigid DNA origami[21] (~ 1 nm) and similar to our earlier reported ~ 5.0 nm resolution for very-low-density lipoprotein (using cryo-EM and IPET)[7], these maps still provided enough information for us to obtain 129 conformations of Bennett linkage. The 129 conformations of Bennett linkages are the first observed dynamic DNA origami in 3D. These structures are all "quadrilateral" shaped but with various conformations ranging from a square-like open conformation to a compacted conformation.

Notably, the IPET 3D density maps were reconstructed from the tilt series with tilt angle range of ± 45°. The limited tilt angle range was responsible for missing wedge of information in Fourier space. As a result, the final 3D reconstruction often contains certain artifacts, such as elongation, blurring, and distracting caustics[23]. To reduce the effects of the missing wedge, the missing wedge data was estimated via computational algorithms reported by numerous groups[23–26]. For example, a computational approach to fill the missing data in 2D electron crystallography was first reported by Agard and Stroud in the 1980s[27]. Recently, a simple Fourier angular filter to effectively suppress the ray artifacts in the single-axis tilting projection acquisition scheme was reported by Kovacik et al.[24]. A statistical reconstruction method, sequential maximum a posteriori expectation maximization was used to compensate the missing-wedge effects by Ruotsalainen et al.[28]. An iterative compressed-sensing optimized non-uniform fast Fourier transform reconstruction (ICON) for missing-wedge restoration was proposed by Sun et al.[25]. Moreover, a generalized Fourier iterative reconstruction algorithm (GENFIRE) proposed by Miao and colleagues[26] showed a certain capability in reducing the missing wedge artifact to achieve a 3D structure with more isotropic resolution. All those approaches benefited the 3D reconstruction by reducing the missing wedge effects and providing a relatively isotropic resolution 3D density map. In our IPET, a similar missing wedge filling algorithm was also included for 3D reconstruction correction.

In the theoretical model, the conformation of Bennett linkages was characterized by the geometrical constraint from joints that connect adjacent arms of the linkage (Fig. 4a). In order to validate the theoretical model and gain some insights into the effects of these geometrical constraints for improved control of DNA origami conformations, we used kinematic analysis to derive the theoretical internal angles within Bennett linkages (details are provided in the Supporting Information), and then compared the results with the above experimental measurements.

The predicted $\alpha$ and $\beta$ angles from kinematic analysis agree with the measurement when both $\alpha$ and $\beta$ are large (Fig. 4b). This result suggests that the observed Bennett linkages with large $\alpha$ and $\beta$ have conformations as designed, and the conformations are dominantly regulated by geometric constraints from joints. However, the predicted $\alpha$ and $\beta$ do not agree well with the measurement when either $\alpha$ or $\beta$ is small (Fig. 4b), although there was little data at these angles since small $\alpha$ or $\beta$ were rarely observed, suggesting these are high-energy configurations. This disagreement may due to the structure distortion of a joint, such as the joint forming the angle $\alpha$ (Fig. 4f). The joints are all formed by two wedge-shaped DNA bundles connected through a shared edge. The mechanical properties of joints is influenced by the angle of the wedges. The joint formed by two obtuse-angle shaped wedges should be stronger than that formed by two acute-angle shaped wedges. Thus, we believed the joint forming the angle $\alpha$ is easier to be collapsed than that joint forming the angle $\beta$, which is consistent to our observations in Fig. 1d, g as well as that in predicted MD simulations.[18] The collapsed structure of the joint leads to additional contraction of the DNA, which further leads to unexpected constrains on the freedom of the angles, especially for angle $\alpha$. Further optimizations of the joints, especially aiming to increase the stiffness of the sharp connecting region by introducing more cross links to supporting the connecting ends, will be helpful toward improving motion control of the DNA origami mechanisms and nanomachines.

Through IPET 3D reconstruction of individual DNA origami Bennett linkages, we validated their 3D structural diversity. Moreover, the geometric analysis of these reconstructions suggests that, although the conformations of Bennett linkage are in good agreement with the design when the linkages are "open" (have large internal angle $\alpha$ and $\beta$), the linkages are unexpectedly flexible and have different conformations from the design when the linkages are more "closed" (have small $\alpha$ or $\beta$). Based on these results and visualization of Bennett linkage model, we proposed potential approaches for improving the control of Bennett linkage conformation. The approach includes redesigning the DNA sequences near the joint that forms the angle $\alpha$ (the middle panel in Fig. 4f) via introducing additional interactions among the DNA strands to stiffen the structure and prevent the structure from collapsing near this joint. The methodology used for studying 3D structure and 3D motion path of Bennett linkages is capable for studying other type of fabricated DOMs, which will provide us with critical feedback to validate the design hypothesis and optimize the design in future engineering and application of mechanisms.

## Methods

**Preparation of OpNS-EM, cryo-EM and cryo-PS specimens**. The negative-staining specimens of DNA origami Bennett linkage were prepared using OpNS protocol[13–15]. Briefly, DNA origami Bennett linkage was diluted to ~ 2 nM with Tris-Borate-EDTA buffer containing 11 mM MgCl$_2$. An aliquot (~ 4 μl) of diluted sample was then placed on an ultra-thin carbon-coated 200-mesh copper grid (CF200-Cu-UL, Electron Microscopy Sciences, Hatfield, PA, USA; Cu-200CN, Pacific Grid-Tech, San Francisco, CA), which had been glow discharged for 15 s. After 1 min incubation, the excessive solution on grid was blotted with filter paper. The grid was then washed with water and stained with 1% (w/v) uranyl formate before air-drying with nitrogen. The cryo-PS specimens were prepared following the procedure described by Zhang et al.[16]. Briefly, an aliquot (~ 4 μl) of DNA origami Bennett linkage sample (concentration ~ 4 nM) was placed on a glow-discharged lacey carbon film-coated copper grid (LC200-Cu, Electron Microscopy Sciences; Cu-200LN, Pacific Grid-Tech) for ~ 1 min. The grid was then washed twice by 1% (w/v) uranyl formate. Instead of air-drying in the last step, the samples were flash-frozen in liquid ethane at ~ 90% humidity and 4 °C with a Leica EM GP rapid-plunging device (Leica, Buffalo Grove, IL, USA) after being blotted with filter paper. The flash-frozen grids were transferred into liquid nitrogen for storage. The cryo-EM specimens were prepared similarly to the cryo-PS specimen, but were not washed by uranyl formate before flash-frozen in liquid nitrogen as described before.[14,16]

**OpNS EM untilted data acquisition and image processing**. OpNS samples were imaged using a Zeiss Libra 120 Plus TEM (Carl Zeiss NTS) operated at 120 kV high tension with a 20 eV energy filter. The untilt micrographs were acquired at Scherzer defocus and with the dose in the range of ~ 10–25 e$^-$ Å$^{-2}$ at a magnification of 50 kx (each pixel of the micrographs corresponds to 0.24 nm in specimen), or a dose of ~ 60–90 e$^-$ Å$^{-2}$ under a magnification of 125 kx (each pixel of the micrographs corresponds to 0.096 nm in specimens) using a Gatan UltraScan 4 K × 4 K charge-coupled device (CCD). The acquired micrographs were Gaussian high-pass filtered to 150 nm and low-pass filtered to 1 nm after the X-ray speckles were removed.

**Cryo-PS data acquisition and image pre-process**. The cryo tilt series were collected by using a FEI Tecnai TF20 TEM (five series) and a Zeiss Libra 120 Plus TEM (15 series). On the FEI Tecnai TF20 TEM (operated at 200 kV), the tilt series were collected from − 48 ° to + 48 ° at 1.5 ° increment using UCSF tomography software package[29] and a Gatan K2 Summit direct electron detection camera at ~ 1 μm defocus and a magnification of 19 kx. At this magnification, the pixel size of the micrographs is 0.185 nm. The electron dose per tilt series is ~50 e$^-$ Å$^{-2}$. On Zeiss Libra 120 Plus TEM (operated at 120 kV), the tilt series were collected from − 45 ° to + 45 ° at 1.5 ° increment using Gatan tomography software and Gatan UltraScan 4 K × 4 K CCD and fully mechanically controlled automated ET software[30] at ~ 1.5 μm defocus and a magnification of 50 kx. At this magnification the pixel size of the micrographs is 0.24 nm. The electron dose per tilt series is from ~ 40 to ~ 180 e$^-$ Å$^{-2}$.

All tilt series were initially aligned using the IMOD[31] software package. CTF of tilt series collected on the FEI Tecnai TF20 TEM was corrected using TomoCTF[32]. The tilt series of the particles in square windows of ~ 95 nm × 95 nm were semi-automatically tracked, windowed using IPET software[5], and finally binned by two times to reduce computation time in subsequent reconstruction.

**IPET 3D reconstruction**. In the pipeline of IPET reconstruction[5], a tilt series containing a single DNA origami Bennett linkage particle was extracted from the full-size tilt series. This allows us to perform "focused" 3D reconstruction, such that the reconstruction is less sensitive to image distortion, tilt-axis variation with respect to tilt angle, and tilt angle offset. In this process, an ab initio 3D density map was directly back-projected in Fourier space and served as the initial model. The refinement was then iteratively invoked to translationally align each tilted particle image to the computed projection. During the refinement, automatically generated Gaussian low-pass filters, soft-boundary circular masks and particle-shaped soft-boundary masks were sequentially applied to the tilt images and references to increase the alignment accuracy[5]. An improved model was then reconstructed based upon the refined alignment at the end of each refinement. The 3D map was then reconstructed by back-projection of filtered and masked particle tilt series. The back projection was performed in Fourier space without weighting. Post 3D reconstruction processing was conducted using our newly developed interactive algorithm to estimate the data in the missing wedge. The resolution was determined based on the FSC that was calculated based upon two reconstructions that were generated using odd and even halves of aligned particle tilt series. It is noteworthy that the generation of the FSC curve did not take into account the missing wedge, as the missing data were estimated in our approach. The frequency at which the FSC curve first falls to a value of 0.5 was used to represent the resolution of the IPET 3D density map. We used the 0.5 FSC criterion, because this method, used for determining the resolution of single particle reconstruction[33], gave a lower resolution than other criteria. All of the IPET density maps in the presented figures were low-pass filtered to 8 nm.

**Conformations of DNA origami Bennett linkage**. Although the resolution (~ 10 nm) of IPET 3D reconstructions of DNA origami Bennett linkage is not sufficient to determine the secondary structure of each individual Bennett linkage, it is sufficient to shed some light on the domain orientations and positions. This information is quite useful in revealing the structural heterogeneous and dynamics. CanDo[17,19] was used to build an initial Bennett linkage model. This model was used to show the flexibility in linkage by flexibly docking it into reconstruction. During this process, each domain (arm) of DNA origami Bennett linkage model was separately translated and rotated to get its best fit to target position in the density map with a minimal overlap among domains. As a result, the achieved conformation of DNA origami Bennett linkage had the same domain structure as initial structure, but differed in the relative position and orientation.

**Statistical analysis of DNA origami Bennett linkage conformations**. To determine the conformational flexibility of DNA origami Bennett linkage, all Bennett linkage conformations were aligned based on their centers using visual molecular dynamics (VMD). The internal angle and dihedral angles of Bennett linkage were then measured and statistically analyzed to explore the structural dynamics of Bennett linkage.

To measure the internal angles between the arms of Bennett linkage, four vectors representing the orientations of arms were included. The angles between vectors were then directly measured as the internal angles. Moreover, based on the four vectors, we also calculated the normal directions of planes formed by two adjacent arms. The angles between normal directions were then measured to

determine the dihedral angles between planes. The obtained internal angles and dihedral angles, and their corresponding histograms were plotted and fitted with a sixth-degree polynomial function using MATLAB. These histograms represent the conformational space of Bennett linkage, thus can be used to explore the conformational flexibility of Bennett linkage.

The data sets generated during and/or analyses during the current study are available from the corresponding author on reasonable request.

**Data availability**. The datasets generated during and/or analyses during the current study are available from the corresponding author on reasonable request. The TEM 3D density maps of 129 DNA origami Bennett linkage are available from the EM data bank (from EMD-7155 to EMD-7285). Detail of these 129 density maps and their corresponding EMDB deposition number is listed in Supplementary Table 1.

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

## Acknowledgements

We thank Drs. Shawn Zheng, Michael Braunfeld, and David Agard at University of California, San Francisco, for their great supporting in cryo-EM data acquisition, K2 images alignment, and editing in the manuscript. This material is based upon work supported by the National Science Foundation under Grant DMR-1344290. Work at the Molecular Foundry was supported by the Office of Science, Office of Basic Energy Sciences of the U.S. Department of Energy under Contract Number DE-AC02-05CH11231. G.R. is supported by the National Heart, Lung, and Blood Institute of the National Institutes of Health (number R01HL115153) and the National Institute of General Medical Sciences of the National Institutes of Health (number R01GM104427). C.E.C., C.H., A.M., H.-J.S., and L.Z. acknowledge the support of National Science Foundation (grant number CMMI-1536862).

## Author contributions

This project was initiated and designed by C.E.C., H.-J.S., and G.R. A.E.M. prepared the DNA origami sample. D.L. and J.L. prepared the TEM samples. D.L., J.L., and G.R. acquired the data. D.L. and G.R. processed the data and solved the IPET 3D structures. D.L. docked and analyzed the models. D.L., A.E.M., J.L., C.-M.H., L.Z., C.E.C., H.-J.S., and G.R. interpreted and manipulated the structures. D.L. drafted the initial manuscript, which was revised by G.R., A.E.M., J.L., C.-M.H., L.Z., C.E.C., and H.-J.S.

## Additional information

**Competing interests:** The authors declare no competing financial interests.

