## [Peer Review File · Nature Communications]

Reviewers' comments:

Reviewer #1 (Remarks to the Author):

This manuscript describes the application of a cryo-EM topography method for characterization of DNA origami nanostructures. The authors use a previously described and characterized design of the Bennett linkage nanostructure to carry out a proof-of-principle reconstruction of its three-dimensional conformation. The authors explored several approaches to obtaining 3D images, arriving at a protocol that gave 16 independent 3D structural models at a 10 nm resolution. The authors used targeted molecular dynamics to obtain atomic models consistent with the 3D electron density maps. Finally, the authors compared the results of the reconstruction to a theoretical model of the mechanism, finding the model to be somewhat incorrect, as it does not properly account for the steric constraints of the design.

Overall, this is an interesting study that describes the first 3D reconstruction of individual DNA origami nanostructures. That alone is, in principle, an impressive achievement. Unfortunately, the 10 nm resolution of the structural reconstructions and the low throughput of the method (only 16 conformations were characterized) are not all that impressive. Consequently, we do not learn much about the Bennett linkage mechanism that we did not know before. Perhaps using a more complex and less characterized object for this proof-of-principle study would have increased the impact of the study on the DNA origami field.

Reviewer #2 (Remarks to the Author):

Scaffolded DNA origami is the only means by which complex nanometer-scale structures can be designed and synthesized with high fidelity for diverse applications in nanoscale science and technology.

While the vast majority of published DNA origami objects are rigid, adopting a single target structure, the Castro lab together with several others have pioneered in recent years conformationally dynamic or polymorphic objects that may adopt vastly different 3D conformations in solution, yet with prescribed mechanisms of conformational change. However, because TEM and AFM imaging are the most common approaches to visualizing DNA origami experimentally, quantitative insight into the 3D conformational changes of these polymorphic objects has been limited.

Here, the authors use 3D single-particle cryo-EM imaging and reconstruction to perform a detailed 3D structural investigation of the Bennett linkage designed and synthesized originally by the Castro lab. This structure can undergo dramatic conformational rearrangements that are visualized and quantified here in 3D for the first time, for any DNA origami object. As such, this study is a landmark structural investigation of the highly dynamic nanoscale reorganization that programmed DNA assemblies can undergo.

Comments the authors should consider in their revision include:

- A schematic introducing the Bennett linkage, even as a simple inset or panel, would be helpful in Figure 1 to orient the reader at the outset of the article
- Labels would ideally be added directly to the figure panels in 3A and B and C to explain how the images are organized, left to right, as well as ideally top and bottom rows

- It is difficult to see the variety of molecular conformations in Figure 3D, these should ideally be separated, or clustered, or other visualization employed to be more useful to the reader
- What is the significance of the Gaussian model fits, or interpretation, if any, in Figure 4? What was learned? Comments in the text and/or figure would be helpful in this regard
- The citation to CanDo should include the Bathe lab's NAR 2012 since the model predictions are not included in the original NM 2011 paper that introduced CanDo using a highly limited set of modeling features (see the CanDo website for its suggested scholarly attributions)

Reviewer #3 (Remarks to the Author):

This manuscript reports the 3D structural dynamics of DNA origami mechanisms using individual-particle electron tomography. By acquiring a cryo tilt series from -48° to $+48^\circ$ at 1.5° increment, the authors reconstruct the 3D structure of 16 DNA origami Bennett linkages with different conformations at ~ 10 nm resolution. Statistical analysis of these conformational changes has led to the understanding of the 3D structural dynamics of Bennett linkage mechanisms. Although a resolution of 10 nm is relatively lower in cryo electron tomography, to my knowledge, this work represents the first observation of 3D structural dynamics of DNA origami. Thus, I think this manuscript will be of interest to the general reader of Nature Communications. However, there are several issues in the current version of the manuscript that need to be carefully addressed before it can be considered for publication.

1. The missing wedge of the tilt series ($\pm 42^\circ$) is large. It is not clear to me why the authors did not acquire the tilt series using a larger tilt range. This can be realized in most electron microscopes. Furthermore, it states in the Methods section, "the generation of the FSC curve did not take into account of missing wedge since the missing data were estimated in our approach." So what is the resolution in the missing wedge direction? How were the missing data estimated in their approach? These details need to be clarified in a revised manuscript.
2. Figures 5a and b show some discrepancy between the measurement and kinematic analysis. What is the error bar in the measurement of the alpha and beta angles? How does the large missing wedge affect the alpha and beta measurements? These need to be clarified in a revised manuscript.
3. Individual-particle electron tomography (IPEM) reconstructs the 3D structure by using the back-projection of filtered and masked particle tilt series. It is well known that the back-projection method is not accurate when there is a limited number of projections or a large missing wedge. For example, a Fourier based iterative algorithm, termed GENFIRE, has been developed for high-resolution 3D reconstruction from a limited number of projections with a missing wedge [Nature 542, 75-79 (2017); Sci. Rep. 7, 10409 (2017)]. It has been demonstrated that GENFIRE produces superior results relative to several other popular tomographic reconstruction techniques including the back-projection method. Furthermore, the algorithm also incorporates angular refinement to reduce the tilt angle error. I suggest the authors to briefly discuss GENFIRE and IPEM in the revised manuscript. I think incorporating GENFIRE may improve the IPEM reconstruction.

Reviewers' comments:

Reviewer #1

Comment #1.1: Overall, this is an interesting study that describes the first 3D reconstruction of individual DNA origami nanostructures. That alone is, in principle, an impressive achievement. Unfortunately, the 10 nm resolution of the structural reconstructions and the low throughput of the method (only 16 conformations were characterized) are not all that impressive. Consequently, we do not learn much about the Bennett linkage mechanism that we did not know before. Perhaps using a more complex and less characterized object for this proof-of-principle study would have increased the impact of the study on the DNA origami field.

Response: In response to the referee's question, we repeated the experiment in last two months, and reconstructed 113 more density map from each individual particles. The resolutions of these 3D reconstructions were little better than before, in which, ~25% of resolutions were better than 7 nm. The statistical analyses and the discussion were updated. The 3D maps of those 129 density maps showed in Fig. 3 as below,

Figure 3 | 3D structures and conformations of 129 individual particles of DNA origami Bennett linkage. (A) The projections of the final 3D reconstructions (with circular mask only) of 4 individual representative particles. (B) The final 3D reconstructions of 4 individual particles of Bennett linkage. (C) Flexibly docking the model of Bennett linkage (yellow ribbon) into the envelopes of 3D density maps. (D) 120 additional density maps from 120 individual particles (orientations are same as that in specimen). (E) Four representative conformations from four individual particles. (F) The corresponding models showed the conformational variety. (G) Aligned all 129 models to each other, and colorized based on the root sum square of angles (angle definition shown in Fig. 4). Scale bars are 20 nm.

A table of the list of all parameters of these 120 density maps was also included as below,

#	EMDB# ¹	TEM ²	CCD ³	Mag. ³	Apix ⁴ (Å)	Dose/img. ⁵ (e-/Å ²)	Dose/set ⁶ (e-/Å ²)	Acq. angle range ⁷	Total img. ⁸	Reconst. angle range ⁹	Cont. ¹⁰	Resol. ¹¹ (nm)	α (°)	β (°)	φ (°)	Geo. mean of ang. ¹² (°)	Fig. ¹³	
1	EMD-7155	Zeiss 120	UltraScan	50 kX	4.8	2.01	122.56	-45° to +45°	61	-45° to 45°	0.299	6.41	101.8	66.4	136.2	119.7	181.3	Fig S2
2	EMD-7156	Zeiss 120	UltraScan	50 kX	4.8	1.69	103.14	-45° to +45°	61	-45° to 45°	0.311	6.29	118.2	59.8	162.2	151.8	222.1	Fig S2
3	EMD-7157	Zeiss 120	UltraScan	50 kX	4.8	1.73	105.58	-45° to +45°	61	-45° to 45°	0.335	10.23	108.2	70.9	162.0	161.1	228.5	Fig S3
4	EMD-7165	Zeiss 120	UltraScan	50 kX	4.8	1.91	116.34	-45° to +45°	61	-45° to 45°	0.334	9.88	104.9	75.5	162.1	157.6	226.1	Fig S3
5	EMD-7170	Zeiss 120	UltraScan	50 kX	4.8	1.65	100.85	-45° to +45°	61	-45° to 45°	0.322	10.94	113.8	62.0	154.2	141.9	209.5	Fig S4
6	EMD-7175	Zeiss 120	UltraScan	50 kX	4.8	1.74	106.18	-45° to +45°	61	-45° to 45°	0.308	6.33	108.3	47.3	122.3	85.4	149.2	Fig S4
7	EMD-7181	Zeiss 120	UltraScan	50 kX	4.8	2.85	173.72	-45° to +45°	61	-45° to 45°	0.261	6.15	127.7	35.4	137.4	85.6	161.9	Fig S5
8	EMD-7184	Zeiss 120	UltraScan	50 kX	4.8	2.79	170.32	-45° to +45°	61	-45° to 45°	0.334	6.83	118.0	61.5	170.4	161.2	234.6	Fig S5
9	EMD-7189	Zeiss 120	UltraScan	50 kX	4.8	2.75	167.67	-45° to +45°	61	-45° to 45°	0.402	6.05	102.3	41.4	107.9	68.2	127.7	Fig S6
10	EMD-7193	Zeiss 120	UltraScan	50 kX	4.8	2.31	140.96	-45° to +45°	61	-45° to 45°	0.275	6.06	101.9	63.3	131.2	112.8	173.0	Fig S6
11	EMD-7196	Zeiss 120	UltraScan	50 kX	4.8	2.83	172.61	-45° to +45°	61	-45° to 45°	0.342	5.94	102.6	46.9	115.7	75.8	138.3	Fig S7
12	EMD-7200	Zeiss 120	UltraScan	50 kX	4.8	2.02	123.34	-45° to +45°	61	-45° to 45°	0.299	6.47	115.8	54.7	146.2	114.7	185.9	Fig S7
13	EMD-7201	Zeiss 120	UltraScan	50 kX	4.8	0.97	58.92	-45° to +45°	61	-42° to 42°	0.310	9.25	105.0	57.9	133.7	104.7	169.8	Fig S8
14	EMD-7203	Zeiss 120	UltraScan	50 kX	4.8	3.00	183.00	-45° to +45°	61	-45° to 45°	0.306	6.65	107.6	68.3	152.6	142.4	208.8	Fig S8
15	EMD-7205	Zeiss 120	UltraScan	50 kX	4.8	2.86	174.22	-45° to +45°	61	-45° to 45°	0.280	5.72	108.6	69.9	163.9	156.3	226.5	Fig S9
16	EMD-7207	Zeiss 120	UltraScan	50 kX	4.8	1.64	100.27	-45° to +45°	61	-45° to 45°	0.321	6.47	126.3	50.6	159.9	140.8	213.0	Fig S9
17	EMD-7210	Zeiss 120	UltraScan	50 kX	4.8	1.68	102.41	-45° to +45°	61	-45° to 45°	0.292	6.36	116.8	61.2	161.8	149.6	220.4	Fig S10
18	EMD-7214	Zeiss 120	UltraScan	50 kX	4.8	1.55	91.85	-45° to +45°	61	-45° to 45°	0.304	8.07	137.3	34.6	168.0	100.0	170.4	Fig S10
...																		
127	EMD-7179	FEI TF20	K2 Summit	19 kX	3.7	0.65	42.13	-48° to +48°	65	-48° to 48°	0.009	12.32	75.7	56.6	90.7	71.6	115.5	Fig S65
128	EMD-7183	FEI TF20	K2 Summit	19 kX	3.7	0.64	41.31	-48° to +48°	65	-48° to 48°	0.012	10.01	57.4	18.6	69.4	6.0	69.6	Fig S65
129	EMD-7186	Zeiss 120	UltraScan	50 kX	4.8	1.89	115.44	-45° to +45°	61	-45° to 45°	0.339	8.48	84.8	51.6	124.5	71.0	143.3	Fig S66

Supplemental Table 1. The parameters used in each IPET 3D reconstructions.

¹ EMDB Index: <https://www.ebi.ac.uk/pdbe/emdb/>

² TEM model: FEI TF20 for FEI TF200 TEM; Zeiss 120 for Zeiss Libra 120 Plus TEM

³ CCD: K2 Summit for Gatan K2 Summit Direct Detector; UltraScan for Gatan UltraScan 4000 4Kx4K CCD

⁴ Angstrom per pixel

⁵ Dose used for each CCD frame

⁶ Dose used for tilt series

⁷ Data acquisition angle range

⁸ Total images in the tilt series

⁹ Reconstruction angle range

¹⁰ Contour used for display

¹¹ 3D reconstruction resolution

¹² Geometric mean of angle θ and φ, i.e. $(\theta^2 + \phi^2)^{1/2}$

¹³ The process of IPET 3D reconstruction showed in supporting Figure

Reviewer #2

Comment #2.1: A schematic introducing the Bennett linkage, even as a simple inset or panel, would be helpful in Figure 1 to orient the reader at the outset of the article

Response: We thank referee for this comment. We added the schematic for the Bennett linkage in Fig. 1A as below,

Figure 1 | OpNS-EM and cryo-PS images of DNA origami Bennett linkage. (A) A schematic introducing the Bennett linkage. (B) A survey OpNS image of DNA origami Bennett linkage. (C) 24 representative particles by OpNS. (D) Zoom-in OpNS image of a representative particle. (E) A survey cryo-PS image of DNA origami Bennett linkage. (F) 24 representative particles by cryo-PS. (G) Zoom-in cryo-PS image of a representative particle. Scale bars are 50 nm in B, C, E and F, and are 20 nm in D and G.

Comment #2.2: Labels would ideally be added directly to the figure panels in 3A and B and C to explain how the images are organized, left to right, as well as ideally top and bottom rows

Response: In response to the comment, we revised Fig 3 and its figure legend as below,

Figure 3 | 3D structures and conformations of 129 individual particles of DNA origami Bennett linkage. (A) The projections of the final 3D reconstructions (with circular mask only) of 4 individual representative particles. (B) The final 3D reconstructions of 4 individual particles of Bennett linkage. (C) Flexibly docking the model of Bennett linkage (yellow ribbon) into the envelopes of 3D density maps. (D) 120 additional density maps from 120 individual particles (orientations are same as that in specimen). (E) Four representative conformations from four individual particles. (F) The corresponding models showed the conformational variety. (G) Aligned all 129 models to each other, and colorized based on the root sum square of angles (angle definition shown in Fig. 4). Scale bars are 20 nm.

Comment #2.3: *It is difficult to see the variety of molecular conformations in Figure 3D, these should ideally be separated, or clustered, or other visualization employed to be more useful to the reader*

Response: In response to the comment, we updated the variety of the molecular conformation based on the root sum square of two internal angles as shown in above response to Comment #2.2.

Comment #2.4: *What is the significance of the Gaussian model fits, or interpretation, if any, in Figure 4? What was learned? Comments in the text and/or figure would be helpful in this regard*

Response: To response to the referee's comment, we revised the Fig. 4 and its legend as below.

Figure 4 | Angle distributions of the 129 particles of DNA origami Bennett linkage. (A) Schematic shows the definition of Bennett linkage internal angles α and β (between two adjacent arms), and the dihedral angles, θ and ϕ . (B) Distribution of internal angle α against β . The pink curve was predicted by using kinematic analysis. (C) The histogram of internal angles α and β was normalized and then fitted using a sixth degree polynomial function, in which, R-factor is 0.992 and STD is 1.28 for α , and R-factor is 0.976 and STD is 3.90 for β . (D) Distribution of the dihedral angle θ against ϕ . The pink curve was predicted by using kinematic analysis. (E) The histogram of dihedral angles θ and ϕ was normalized and then fitted using a sixth degree polynomial function, in which, R-factor is 0.997 and STD is 1.36 for α , and R-factor is 0.937 and STD is 4.42 for β . (F) Three orientations of the model showed the interaction regions between two adjacent arms that formed the internal angle α and β .

Comment #2.5: The citation to CanDo should include the Bathe lab's NAR 2012 since the model predictions are not included in the original NM 2011 paper that introduced CanDo using a highly limited set of modeling features (see the CanDo website for its suggested scholarly attributions)

Response: To response to the referee's comment, we revised the following sentences by including the discussion with the predicted result from CanDo as below,

Line 93:

However, the arms have a curved shape (Fig. 1D) which was not predicted in designed DNA origami model, suggesting the structures have substantial flexibility. The flexible arms observed was consistent with that from the computational prediction¹⁷ and simulations¹⁸. Moreover, the clear shape of arms and overall particle shapes suggest those images were useful for revealing the basic structure of DNA origami Bennett linkage.

Line 114:

However, the curvy arms (Fig. 1G) confirmed the flexibility of DNA bundles observed from the OpNS, and further supported the prediction based on the computation and molecular dynamic simulations^{17,18}. Moreover, the high contrast of images with less influence of orientation or flatness makes it possible to perform 3D reconstruction from each individual particle of DNA origami Bennett linkage.

Line 135:

Although the resolution of reconstructed map was insufficient to determine the secondary structure of this Bennett linkage, the overall map can be used as a constraint to flexibly dock an initial model of the Bennett linkage model, built using CanDo,^{17,19} to obtain a new Bennett linkage conformation. By

satisfying both the best overall fit to the density map and a minimal overlap among arms within model, we changed the joint angles between two adhered arms (Fig. 2D) to obtain a new conformation of the Bennett linkage that differs from the original model (Fig. 2D).

Line 340:

Conformations of DNA origami Bennett linkage

While the resolution (~10 nm) of IPET 3D reconstructions of DNA origami Bennett linkage is not sufficient to determine the secondary structure of each individual Bennett linkage, they are sufficient to shed some light on the domain orientations and positions. This information is quite useful to reveal the structural heterogeneous and dynamics. CanDo^{17,19} was used to build an initial Bennett linkage model. This model was used to show the flexibility in linkage by flexibly docking it into reconstruction. During this process, each domain (arm) of DNA origami Bennett linkage model was separately translated and rotated to get its best fit to target position in the density map with a minimal overlap among domains. As a result, the achieved conformation of DNA origami Bennett linkage had the same domain structure as initial structure, but differed in the relative position and orientation.

Reviewer #3

Comment #3.1: *The missing wedge of the tilt series ($\pm 42^\circ$) is large. It is not clear to me why the authors did not acquire the tilt series using a larger tilt range. This can be realized in most electron microscopes. Furthermore, it states in the Methods section, “the generation of the FSC curve did not take into account of missing wedge since the missing data were estimated in our approach.” So what is the resolution in the missing wedge direction? How were the missing data estimated in their approach? These details need to be clarified in a revised manuscript.*

Response: In response to the referee’s question, we have added a paragraph to page 5 as below:

Line 224:

Notably, the IPET 3D density maps were reconstructed from the tilt series with tilt angle range of $\pm 45^\circ$, in which the missing wedge is large. To reduce the effects of the missing wedge, the missing wedge data were estimated via our developed computational algorithm coded in our IPET software package (manuscript is in preparing). The various approaches to compute the missing data have been reported by numerous groups. For example, a computational approach to fill the missing data in 2D electron crystallography was first reported by Agard and Stroud in the 1980s.²³ Recently, a simple Fourier angular filter to effectively suppresses the ray artifacts in the single-axis tilting projection acquisition scheme was reported by Kovacic, *et. al.*²⁴. A statistical reconstruction method, sequential maximum a posteriori expectation maximization (sMAP-EM) was used to compensate the missing-wedge effects by Ruotsalainen *et. al.*²⁵. An iterative compressed-sensing optimized non-uniform fast Fourier transform reconstruction (ICON) for missing-wedge restoration was proposed by Sun *et. al.*²⁶. Moreover, a generalized Fourier iterative reconstruction algorithm (GENFIRE) proposed by Miao *et. al.* showed a certain capability in reducing the missing wedge artifact to achieve a 3D structure with more isotropic resolution.²⁷ All those approaches benefited the 3D reconstruction by reducing the missing wedge effects and providing a relatively anisotropic resolution 3D density map. Thus, the angles measured from the centers of the arms within those corrected 3Ds have limited influence from the missing wedge.

Comment #3.2: *Figures 5a and b show some discrepancy between the measurement and kinematic analysis. What is the error bar in the measurement of the alpha and beta angles? How does the large missing wedge affect the alpha and beta measurements? These need to be clarified in a revised manuscript.*

Response: In response to this question, the second paragraph of page 10 has been revised as follows:

Figure 4 | Angle distributions of the 129 particles of DNA origami Bennett linkage. (A) Schematic shows the definition of Bennett linkage internal angles α and β (between two adjacent arms), and the dihedral angles, θ and ϕ . (B) Distribution of internal angle α against β . The pink curve was predicted by using kinematic analysis. (C) The histogram of internal angles α and β was normalized and then fitted using a sixth degree polynomial function, in which, R-factor is 0.992 and STD is 1.28 for α , and R-factor is 0.976 and STD is 3.90 for β . (D) Distribution of the dihedral angle θ against ϕ . The pink curve was predicted by using kinematic analysis. (E) The histogram of dihedral angles θ and ϕ was normalized and then fitted using a sixth degree polynomial function, in which, R-factor is 0.997 and STD is 1.36 for α , and R-factor is 0.937 and STD is 4.42 for β . (F) Three orientations of the model showed the interaction regions between two adjacent arms that formed the internal angle α and β .

We also added following paragraph to discuss the missing wedge effects as below,

Line 224:

Notably, the IPET 3D density maps were reconstructed from the tilt series with tilt angle range of $\pm 45^\circ$, in which the missing wedge is large. To reduce the effects of the missing wedge, the missing wedge data were estimated via our developed computational algorithm coded in our IPET software package (manuscript is in preparing). The various approaches to compute the missing data have been reported by numerous groups. For example, a computational approach to fill the missing data in 2D electron crystallography was first reported by Agard and Stroud in the 1980s.²³ Recently, a simple Fourier angular filter to effectively suppresses the ray artifacts in the single-axis tilting projection acquisition scheme was reported by Kovacic, *et. al.*²⁴ A statistical reconstruction method, sequential maximum a posteriori expectation maximization (sMAP-EM) was used to compensate the missing-wedge effects by Ruotsalainen *et. al.*²⁵. An iterative compressed-sensing optimized non-uniform fast Fourier transform reconstruction (ICON) for missing-wedge restoration was proposed by Sun *et. al.*²⁶. Moreover, a generalized Fourier iterative reconstruction algorithm (GENFIRE) proposed by Miao *et. al.* showed a certain capability in reducing the missing wedge artifact to achieve a 3D structure with more isotropic resolution.²⁷ All those approaches benefited the 3D reconstruction by reducing the missing wedge effects and providing a relatively anisotropic resolution 3D density map. Thus, the angles measured from the centers of the arms within those corrected 3Ds have limited influence from the missing wedge.

Comment #3.3: Individual-particle electron tomography (IPET) reconstructs the 3D structure by using the back-projection of filtered and masked particle tilt series. It is well known that the back-projection method is not accurate when there is a limited number of projections or a large

missing wedge. For example, a Fourier based iterative algorithm, termed GENFIRE, has been developed for high-resolution 3D reconstruction from a limited number of projections with a missing wedge [Nature 542, 75-79 (2017); Sci. Rep. 7, 10409 (2017)]. It has been demonstrated that GENFIRE produces superior results relative to several other popular tomographic reconstruction techniques including the back-projection method. Furthermore, the algorithm also incorporates angular refinement to reduce the tilt angle error. I suggest the authors to briefly discuss GENFIRE and IPEM in the revised manuscript. I think incorporating GENFIRE may improve the IPEM reconstruction.

Response: This comment is essentially similar to the *Comment #3.1*, in which, the discussion about GENFIRE with other approaches has been stated briefly as following.

Line 224:

Notably, the IPET 3D density maps were reconstructed from the tilt series with tilt angle range of $\pm 45^\circ$, in which the missing wedge is large. To reduce the effects of the missing wedge, the missing wedge data were estimated via our developed computational algorithm coded in our IPET software package (manuscript is in preparing). The various approaches to compute the missing data have been reported by numerous groups. For example, a computational approach to fill the missing data in 2D electron crystallography was first reported by Agard and Stroud in the 1980s.²³ Recently, a simple Fourier angular filter to effectively suppresses the ray artifacts in the single-axis tilting projection acquisition scheme was reported by Kovacic, *et. al.*²⁴. A statistical reconstruction method, sequential maximum a posteriori expectation maximization (sMAP-EM) was used to compensate the missing-wedge effects by Ruotsalainen *et. al.*²⁵. An iterative compressed-sensing optimized non-uniform fast Fourier transform reconstruction (ICON) for missing-wedge restoration was proposed by Sun *et. al.*²⁶. Moreover, a generalized Fourier iterative reconstruction algorithm (GENFIRE) proposed by Miao *et. al.* showed a certain capability in reducing the missing wedge artifact to achieve a 3D structure with more isotropic resolution.²⁷ All those approaches benefited the 3D reconstruction by reducing the missing wedge effects and providing a relatively anisotropic resolution 3D density map. Thus, the angles measured from the centers of the arms within those corrected 3Ds have limited influence from the missing wedge.

REVIEWERS' COMMENTS:

Reviewer #1 (Remarks to the Author):

In the revised version of the manuscript, the authors present additional 113 single particle reconstructions of the Bennett linkage, bringing the total number of such reconstructions to 129. This is an impressive accomplishment that will serve as a strong endorsement for the application of the single particle reconstruction technique to the DNA origami field. Furthermore, the 129 TEM 3D density maps obtained by the study will provide an invaluable data set for future refinement of computational models of DNA origami systems.

Nevertheless, the present version of the manuscript has several problems.

1. The Authors added new text on lines 93-95 and 114-116, which implies that flexibility of the bundles seen in cryo images was already predicted by the CanDO program back in 2012 (Ref 17) and by the OxDNA model (Ref 18). The authors should either present data supporting this claim (such as explicit comparison between the results of CanDO and OxDNA simulations with the results of 3D reconstructions) or revise the sentences to adequately reflect the degree of foresight and the degree of agreement between the simulations and experiment.

2. Almost all newly added sentences have grammar, spelling or logic problems, starting with the title (the linkage is not capitalized). It looks like the new text was never properly edited. Below, I present several instances of bad writing, which are only examples: there are problems in almost every new sentence (and in several old sentences as well)

Lines 94-95: "The flexible arms observed was consistent with": grammar problem.

Line 96: "Moreover, the clear shape of arms and overall particle shapes suggest " What does a "clear shape of arms" mean? The arms are transparent?

Line 252: "This disagreement was caused by the missed observed data, suggesting that the Bennett linkages with small α or β rarely showed." Rarely showed what?

Line 254: "... form the angle, being sharp, especially for β , in which less DNA behind can support ...". DNA does not have a "behind".

Line 255: "The structure of DNA near these sharp connecting regions can be easily crashed by the flexibility of the arm". How can flexibility of the arms crush a structure?

268 "Based on these results and visualization of Bennett linkage model, we proposed potential approaches for improving the control of Bennett linkage conformation." Great! Why don't you describe those potential approaches in the next several sentences?

3. The new paragraph spanning lines 224 through 239 has additional problems. First, the word "wedge" appears out of nowhere in the first sentence. Not every person reading this article is expert in 3D reconstruction. Avoid using jargon. Present your discussion using a language that is accessible to an undergraduate science major. A bigger problem, however, is a revelation that the 3D models presented in the manuscript were constructed using an undocumented algorithm (manuscript in preparation), meaning that the present manuscript does not provide a complete account of the methodological procedures. The authors should either described the algorithm in the SI or deposit a preprint of the manuscript into an archive service so that the procedures used by the authors could be

reproduced by others.

4. Supporting Table 1 is missing from the SI.

Reviewer #2 (Remarks to the Author):

The authors have addressed my comments so I am happy to recommend publication of this very interesting work.

Reviewer #3 (Remarks to the Author):

The authors have fully addressed my points in their revised manuscript and I recommend its publication in Nat. Commun.

REVIEWERS' COMMENTS:

Reviewer #1

Comment #1.1. *The Authors added new text on lines 93-95 and 114-116, which implies that flexibility of the bundles seen in cryo images was already predicted by the CanDO program back in 2012 (Ref 17) and by the OxDNA model (Ref 18). The authors should either present data supporting this claim (such as explicit comparison between the results of CanDO and OxDNA simulations with the results of 3D reconstructions) or revise the sentences to adequately reflect the degree of foresight and the degree of agreement between the simulations and experiment.*

Response:

Comment #1.2. *Almost all newly added sentences have grammar, spelling or logic problems, starting with the title (the linkage is not capitalized). It looks like the new text was never properly edited. Below, I present several instances of bad writing, which are only examples: there are problems in almost every new sentence (and in several old sentences as well)*

Response: We thank this referee for these comments, the revised manuscript has been edited by three native English speakers.

Lines 94-95: “The flexible arms observed was consistent with”: grammar problem.

Line 96: “Moreover, the clear shape of arms and overall particle shapes suggest “ What does a “clear shape of arms” mean? The arms are transparent?”

Response: The corresponding sentences have been revised as following,

However, the arms have a curved shape (**Fig. 1d**) which was not predicted in design, suggesting the structures have substantial flexibility. This observation was consistent with that from computational predictions¹⁷ and simulations¹⁸. Moreover, the distinct shape of arms and overall particle shapes suggest those images were useful for revealing the basic structure of a DNA origami Bennett linkage.

Line 252: “This disagreement was caused by the missed observed data, suggesting that the Bennett linkages with small α or β rarely showed.” Rarely showed what?

Line 254: “... form the angle, being sharp, especially for β , in which less DNA behind can support ... “. DNA does not have a “behind”.

Line 255: “The structure of DNA near these sharp connecting regions can be easily crashed by the flexibility of the arm”. How can flexibility of the arms crush a structure?

Response: The corresponding sentences have been revised as following,

This disagreement was caused by missing observational data, suggesting that the Bennett linkages with small α or β were rarely observed. This disagreement may due to the structure collapsing of a joint, such as the joint forming the angle α (**Fig. 4f**). The joints are all formed by two wedge shaped DNA structure and connected through a shared edge. The hardness of joints is defined by the angle of the wedges. The joint formed by two obtuse-angle shaped wedges should be stronger than that formed by two acute-angle shaped wedges. Thus, we believed the joint forming the angle α is easier to be collapsed than that joint forming the angle β , which is consistent to our observations in **Fig. 1d** and **1g** as well as that in the predicted MD simulations.¹⁸

Line 268 “Based on these results and visualization of Bennett linkage model, we proposed potential approaches for improving the control of Bennett linkage conformation.” Great! Why don't you describe those potential approaches in the next several sentences?

Response: The corresponding sentences have been revised as following,

Based on these results and visualization of Bennett linkage model, we proposed potential approaches for improving the control of Bennett linkage conformation. The approach includes redesigning the DNA sequences near the joint that forms the angle α (the middle panel in **Fig. 4f**) via introducing additional interactions among the DNA bases to stiffen the structure and prevent the structure from collapsing near this joint.

Comment #1.3. *The new paragraph spanning lines 224 through 239 has additional problems. First, the word “wedge” appears out of nowhere in the first sentence. Not every person reading this article is expert in 3D reconstruction. Avoid using jargon. Present your discussion using a language that is accessible to an undergraduate science major. A bigger problem, however, is a revelation that the 3D models presented in the manuscript were constructed using an undocumented algorithm (manuscript in preparation), meaning that the present manuscript does not provide a complete account of the methodological procedures. The authors should either described the algorithm in the SI or deposit a preprint of the manuscript into an archive service so that the procedures used by the authors could be reproduced by others.*

Response: To response to referee’s comments, we revised the paragraph as following,

Notably, the IPET 3D density maps were reconstructed from the tilt series with tilt angle range of $\pm 45^\circ$. The limited tilt angle range was responsible for an insidious missing wedge of information in Fourier space. As a result, the final 3D reconstruction often contains certain artifacts, such as elongation, blurring, and distracting caustics.²³ To reduce the effects of the missing wedge, the missing wedge data was estimated via computational algorithms reported by numerous groups.^{23, 24, 25, 26} For example, a computational approach to fill the missing data in 2D electron crystallography was first reported by Agard and Stroud in the 1980s.²⁷ Recently, a simple Fourier angular filter to effectively suppress the ray artifacts in the single-axis tilting projection acquisition scheme was reported by Kovacic, *et. al.*²⁴. A statistical reconstruction method, sequential maximum a posteriori expectation maximization (sMAP-EM) was used to compensate the missing-wedge effects by Ruotsalainen *et. al.*²⁸. An iterative compressed-sensing optimized non-uniform fast Fourier transform reconstruction (ICON) for missing-wedge restoration was proposed by Sun *et. al.*²⁵. Moreover, a generalized Fourier iterative reconstruction algorithm (GENFIRE) proposed by Miao *et. al.* showed a certain capability in reducing the missing wedge artifact to achieve a 3D structure with more isotropic resolution.²⁶ All those approaches benefited the 3D reconstruction by reducing the missing wedge effects and providing a relatively anisotropic resolution 3D density map. In our IPET, a similar missing wedge filling algorithm was also included for 3D reconstruction correction

Comment #1.4. *Supporting Table 1 is missing from the SI.*

Response: We thank this referee for this comment. The supplementary table has been included in the Supplementary Information.

Reviewer #2 (Remarks to the Author):

The authors have addressed my comments so I am happy to recommend publication of this very interesting work.

Response: We thank this referee for this comment.

Reviewer #3 (Remarks to the Author):

The authors have fully addressed my points in their revised manuscript and I recommend its publication in Nat. Commun.

Response: We thank this referee for this comment.